# CALIBRATION-THEN-CALCULATION: A VARIANCE REDUCED METRIC FRAMEWORK

## ABSTRACT

Deep learning has been widely adopted across various fields, but there has been little focus on evaluating the performance of deep learning pipelines. With the increased use of large datasets and complex models, it has become common to run the training process only once and compare the result to previous benchmarks. However, this procedure can lead to imprecise comparisons due to the variance in neural network evaluation metrics. The metric variance comes from the randomness inherent in the training process of deep learning pipelines. Traditional solutions such as running the training process multiple times are usually not feasible in deep learning due to computational limitations. In this paper, we propose a new metric framework, Calibrated Loss, that addresses this issue by reducing the variance in its vanilla counterpart. As a result, the new metric has a higher accuracy to detect effective modeling improvement. Our approach is supported by theoretical justifications and extensive experimental validations in the context of Deep Click-Through Rate Prediction Models and Image Classification Models.

## 1 INTRODUCTION

The progress in machine learning is largely influenced by experimental outcomes, particularly in the era of deep learning. Researchers often evaluate the performance of new methods by comparing them to previous benchmark results, in order to demonstrate their superiority. However, it is well known that the performance of deep learning models can vary greatly, even when using the same pipeline (Picard, 2021), where, in this work, we define the pipeline broadly, which includes but is not limited to the selection of feature sets, model architectures, optimization algorithms, initialization schemes, and hyper-parameters. This variability poses a significant challenge to evaluating deep learning methods because it is non-trivial to determine whether the performance gain is a result of the proposed method or simply due to randomness.

In fact, it has been shown that by selecting a fortunate random initialization seed, one can achieve a model performance that is significantly better than average (Picard, 2021). This difference can be substantial enough to be used as a strong argument for publications in selective venues (Picard, 2021).

This issue is exacerbated in industry where the production model performance is hard to get improvement, while there are hundreds of machine learning engineers working on the same model at the same time. The performance gain of a modeling proposal is usually small and within the metric variance, making it difficult to judge the effectiveness of the modeling proposal.

To address this issue, a common approach is to run the training pipeline multiple times and report the average, standard deviation, minimum, and maximum performance scores (Picard, 2021). However, with the rise of large training data and big models, this approach is not always practical due to limited computational resources (Bouthillier & Varoquaux, 2020).

In this work, we take a different approach to solving this problem by designing a new metric that can evaluate proposed deep learning methods more reliably. This new metric has a smaller variance and achieves more accurate comparisons between different deep learning pipelines. We validate our approach in the context of Deep Click-Through Rate Prediction Models and Image Classification Models.

Our main contributions are:

1) Formulating the deep learning pipeline evaluation problem and proposing to tackle it by designing new metrics.

2) Proposing a new metric framework, Calibrated Loss, which can mitigate the above deep learning pipeline evaluation issue.

3) Conducting extensive experiments to demonstrate the effectiveness of the proposed metric, using synthetic dataset, ads click dataset and image classification dataset.

4) Providing theoretical guarantees under a particular setting of linear regression that the proposed metric has a smaller variance than its vanilla counterpart.

## 2 PRELIMINARIES AND PROBLEM SETTING

In this work, we are examining the standard supervised learning setting, where we assume that the training data and test data are randomly drawn from an unknown distribution in an i.i.d. manner, denoted as $\mathcal{D}$.

Our goal is to develop a good pipeline that maps from a training distribution to a possibly random model, $h \in \mathcal{H}$, that generalizes well during the test time. As we mentioned in Introduction, the pipeline includes the whole procedures of training a model, including the selection of model architectures, optimization algorithms, initialization schemes, and hyper-parameters. Model performance is evaluated by a metric, $e$, and thus the expected performance of a model $h$ during the test time is

$$R_e(h) = \mathbb{E}_{\mathcal{D}}[e(h(X), Y)|h]. \tag{1}$$

In practice, $R_e(h)$ is estimated by finite-sample average on the test dataset $\hat{\mathcal{D}}_{\text{test}}$. That is,

$$\hat{R}_e(h, \hat{\mathcal{D}}_{\text{test}}) = \frac{1}{|\hat{\mathcal{D}}_{\text{test}}|} \sum_{(x,y) \in \hat{\mathcal{D}}_{\text{test}}} e(h(x), y). \tag{2}$$

It's worth noting that the expected risk, $R_e(h)$, is a random variable, as $h$ is random and it depends on a specific model that is produced by the underlying deep learning pipeline. The output model is random due to data randomness from the sample collection and intrinsic randomness during the training process in the deep learning pipeline, such as data order and randomness from stochastic gradient descent. Therefore, a proper evaluation and comparison of different deep learning pipelines should take into account the distribution of $R_e(h)$ (Bouthillier et al., 2021). It's also important to note that the term "deep learning pipeline" in this context is general, as we consider different model configurations (e.g. different model hyperparameters) as different "deep learning pipelines", even though they may belong to the same model class.

To compare the performance of different deep learning pipelines, we should compare the distribution of $R_e(h, \mathcal{D}_{\text{test}})$. When comparing two pipelines, $A$ and $B$, we say that pipeline $A$ is better than pipeline $B$ with respect to metric $e$ if and only if the probability that pipeline $A$ produces a better model (i.e. smaller risk), measured by the metric $e$, is greater than 0.5. This is represented by the inequality:

$$P(R_e(h_A) < R_e(h_B)) > 0.5, \tag{3}$$

where $h_A$ and $h_B$ are random variables representing the output models produced by pipeline $A$ and $B$ respectively.

Our objective is to compare the performance of two pipelines, $A$ and $B$, with respect to the metric $e$ by running the training pipeline only once. Ideally, to estimate $P(R_e(h_A) < R_e(h_B))$, we could use the Monte Carlo method, but this requires a huge amount of computational resources which is not realistic. In this work, we propose a different approach: we aim to come up with an alternative metric that has roughly the same mean but a smaller variance, i.e., for a new metric $e_1$, we would like to have

$$\mathbb{E}[R_{e_1}(h)] \approx \mathbb{E}[R_e(h)] \text{ and } var\left(R_{e_1}(h)\right) < var\left(R_e(h)\right),$$

where the randomness is from the pipeline producing $h$. As a result, the new metric is able to compare the performance of pipelines $A$ and $B$ with limited computational resources more accurately.

Assuming that pipeline $A$ is better than $B$ with respect to the metric $e$ (i.e. pipeline $A$ is more likely to produce a better model than pipeline $B$ in the ground truth if measured by metric $e$), we say that a metric $e_1$ is better than $e_2$ with respect to metric $e$ if and only if the probability that pipeline $A$ produces a better model than pipeline $B$ measured by metric $e_1$ is greater than the probability measured by metric $e_2$. This is represented by the inequality:

$$P(R_{e_1}(h_A) < R_{e_1}(h_B)) > P(R_{e_2}(h_A) < R_{e_2}(h_B)) \tag{4}$$

In other words, using metric $e_1$ is more likely to accurately detect that pipeline $A$ is better than pipeline $B$, which aligns with the ground truth. Here, we allow for a general form of the risk function, which may not admit the expectation form; i.e., $R_{e_1}(h)$ may not necessarily have the form $\mathbb{E}_{\mathcal{D}}[e_1(h(X), Y)]$.

Here, we assume without loss of generality that pipeline $A$ is better than $B$ with respect to the metric $e$. We define the accuracy of a metric $\bar{e}$ with respect to metric $e$ and pipeline A and B as:

$$\text{Acc}(\bar{e}) \triangleq P(R_{\bar{e}}(h_A) < R_{\bar{e}}(h_B)). \tag{5}$$

Our goal is to find a metric $\bar{e}$ associated with the risk function $R_{\bar{e}}$ that has higher accuracy than the original metric $e$ for a wide range of pipelines $A$ and $B$. In the next section, we will present a new metric framework, Calibrated Loss. In the context of Deep Click-Through Rate Prediction Models, a special instance of Calibrated Loss, Calibrated Log Loss achieves higher accuracy than the vanilla Log Loss. The intuition is that the bias in the function $h$ is always volatile and carries on a large amount of randomness. Calibrating the bias will usually not change the comparison between two pipelines but can reduce the randomness. In Section 4, we will present a theoretical analysis that justifies this intuition by showing that our new metric has a smaller variance under the linear regression setting. Through extensive experiments in Section 5, we will show that Calibrated Log Loss achieves higher accuracy than Log Loss for a wide range of pipelines, including those with different batch sizes, number of features, model architectures, regularization weights, model sizes, etc.

## 3 Calibrated Loss Framework

**Calibrated Log Loss (Binary Classification)** In the field of Deep Click-Through Rate Prediction Models, it is common for models to overfit when trained for more than one epoch (Zhou et al., 2018; Zhang et al., 2022). As a result, models are often only trained for a single epoch in practice (Zhang et al., 2022), leaving it uncertain if the model has been fully optimized. This leads to the volatility of the bias term in the final layer of neural networks, creating additional randomness. To address this issue, we propose the following risk function:

$$R_{e_1}(h) = \min_c \mathbb{E}_{\mathcal{D}}[Y \log(h^c(X)) + (1 - Y) \log(1 - h^c(X))|h],$$

where $h^c(X) = (1 + e^{-\text{logit}(h(X))+c})^{-1}$ and $\text{logit}(p) = \log(p/(1-p))$.

To execute the aforementioned procedure on a finite-sample test set, we split the test data $\hat{\mathcal{D}}_{\text{test}}$ into two parts: a test-validation dataset $\hat{\mathcal{D}}_{\text{val}-\text{test}}$ and a remaining test dataset $\hat{\mathcal{D}}_{\text{remaining}-\text{test}}$. By using the test-validation dataset $\hat{\mathcal{D}}_{\text{val}-\text{test}}$, we are able to correct the bias term, and then calculate the Log Loss using $\hat{\mathcal{D}}_{\text{remaining}-\text{test}}$ with the bias-adjusted predictions. This is referred to as Calibrated Log Loss. The calculation procedure is outlined in Algorithm 1.

Mathematically speaking, we define bias-adjusted predictions as: $q_i = h^c(x_i)$ for $x_i$ in the test set where $c$ is the bias-adjusted term we are optimizing.

To optimize $c$, the following optimization program is solved, which is the log loss between bias-adjusted predictions $q_i$ and labels $y_i$:

$$\min_c \left\{ - \sum_{\substack{(x,y) \in \\ \hat{\mathcal{D}}_{\text{val}-\text{test}}}} (y \log(h^c(x)) + (1 - y) \log(1 - h^c(x))) \right\}. \tag{6}$$

It can be easily shown that, after optimization, the bias-adjusted predictions $q_i$ are well-calibrated in $\hat{\mathcal{D}}_{\text{val}-\text{test}}$, meaning that $\sum_{i \in \hat{\mathcal{D}}_{\text{val}-\text{test}}} q_i = \sum_{i \in \hat{\mathcal{D}}_{\text{val}-\text{test}}} y_i$.

Let $c^*$ be the minimizer of the optimization problem (6). The final risk and metrics are

$$\hat{R}_{e_1}(h, \hat{\mathcal{D}}_{\text{test}}) = \frac{1}{|\hat{\mathcal{D}}_{\text{remaining}-\text{test}}|} \sum_{\substack{(x,y) \in \\ \hat{\mathcal{D}}_{\text{remaining}-\text{test}}}} e_1(h(x), y),$$

and

$$e_1(h(x), y) = y \log(h^{c^*}(x)) + (1 - y) \log(1 - h^{c^*}(x)).$$

**Explanations** The optimization problem (6) corrects the bias term of original predictions $h(x)$ using test-validation dataset $\hat{\mathcal{D}}_{\text{val}-\text{test}}$. The bias-adjusted predictions $h^{c^*}(x)$ is guaranteed to be well-calibrated in $\hat{\mathcal{D}}_{\text{val}-\text{test}}$, hence the name Calibrated Log Loss.

---

**Algorithm 1** Calculate Calibrated Log Loss

---

1: **Input:** Model $M$, labeled test data $\hat{\mathcal{D}}_{\text{test}}$.
2: **Output:** Calibrated Log Loss: $\hat{R}_{e_1}(h, \hat{\mathcal{D}}_{\text{test}})$.
3: Partition $\hat{\mathcal{D}}_{\text{test}}$ into $\hat{\mathcal{D}}_{\text{val}-\text{test}}$ and $\hat{\mathcal{D}}_{\text{remaining}-\text{test}}$.
4: Compute model predictions on $\hat{\mathcal{D}}_{\text{val}-\text{test}}$ and $\hat{\mathcal{D}}_{\text{remaining}-\text{test}}$, and obtain the model predictions $p_i^{\text{val}-\text{test}}$ and $p_i^{\text{remaining}-\text{test}}$.
5: Solve the optimization problem (6) using $p_i^{\text{val}-\text{test}}$ and $y_i^{\text{val}-\text{test}}$ and obtain the learned bias term $c^*$.
6: Calculate bias-adjusted predictions $q_i^{\text{remaining}-\text{test}}$ using formula $q_i = h^{c^*}(x_i)$.
7: Calculate the Calibrated Log Loss $\hat{R}_{e_1}(h, \hat{\mathcal{D}}_{\text{test}})$ as the Log Loss of $q_i^{\text{remaining}-\text{test}}$ and $y_i^{\text{remaining}-\text{test}}$.

---

**Generalization to Multiclass Classification** Instead of optimizing problem (6) to calculate calibrated predictions, we use "Temperature Scaling" method proposed in (Guo et al., 2017a). The rest are exactly the same.

**Generalization to Quadratic Loss** Calibrated Quadratic Loss is calculated in a similar manner as Calibrated Log Loss, i.e. first perform calibration on $\hat{\mathcal{D}}_{\text{val}-\text{test}}$ and calculate bias-adjusted predictions on $\hat{\mathcal{D}}_{\text{remaining}-\text{test}}$. Here, we define the Quadratic Loss and Calibrated Quadratic Loss:

$$e(h(x), y) = (y - h(x))^2, \text{ and}$$

$$e_1(h(x), y) = (y - h(x) - (\mathbb{E}_{\mathcal{D}}[Y] - \mathbb{E}_{\mathcal{D}}[h(X)|h]))^2.$$

## 4 Theory on Linear Regression

In this section, we provide theoretical justification that our new metric has a smaller variance than its vanilla counterpart under Linear Regression setting, where the randomness only comes from the data randomness. We choose to provide a theoretical guarantee under Linear Regression due to its simplicity. We empirically verify our method's performance under Logistic Regression and Neural Networks in the next section. Note that in Linear Regression, Quadratic Loss is used instead of Log Loss. As a result, in our theory, we compare the variance of Calibrated Quadratic Loss with vanilla Quadratic Loss.

**Theorem 4.1.** *Suppose that the features $X \in \mathbb{R}^d$ and the label $Y$ are distributed jointly Gaussian. We consider linear regression $h(x) = \beta^\top x + \alpha$. Let $\hat{\beta}_n$ be the coefficient learned from the training data with sample size $n$. Then, we have*

$$\left(1 + \frac{1}{n}\right) \mathbb{E}[e_1(h(X), Y)|\hat{\beta}_n] = \mathbb{E}[e(h(X), Y)|\hat{\beta}_n],$$

*where the expectation is taken over the randomness over both the training and test samples.*

Let $\hat{\alpha}_n$ be the learned intercept. Note that the original risk and the calibrated risk are

$$R_e(h) = \mathbb{E}[e(h(X), Y)|\hat{\beta}_n, \hat{\alpha}_n], \text{ and}$$
$$R_{e_1}(h) = \mathbb{E}[e_1(h(X), Y)|\hat{\beta}_n, \hat{\alpha}_n] = \mathbb{E}[e_1(h(X), Y)|\hat{\beta}_n].$$

Therefore, Theorem 4.1 implies that

$$(1 + 1/n)\mathbb{E}[R_{e_1}(h)] = \mathbb{E}[R_e(h)].$$

Furthermore, to make $e$ and $e_1$ comparable, we should scale $e_1$ to $(1 + 1/n)e_1$. We demonstrate that after scaling, $(1 + 1/n)R_{e_1}(h)$ has a smaller variance than $R_e(h)$ in the next corollary. In practice, as $(1 + 1/n)$ is a constant as long as the training sample size is fixed, we can directly compare two pipelines using $R_{e_1}(h)$.

**Corollary 4.2.** *Suppose that $h_1(x)$ and $h_2(x)$ are two different learned linear functions in different feature sets. Then, we have*

$$\mathbb{E}[R_e(h_1)] = \mathbb{E}[R_e(h_2)] \Leftrightarrow \mathbb{E}[R_{e_1}(h_1)] = \mathbb{E}[R_{e_1}(h_2)] \tag{7}$$

*and*

$$var((1 + 1/n)\, R_{e_1}(h)) < var(R_e(h))$$

*for any $h$ learned from linear regression.*

Corollary 4.2 indicates that Calibratied Quadratic Loss has a smaller variance than vanilla Quadratic Loss without changing the mean after appropriate scaling. Note that smaller variance and higher accuracy (Inequality 4) are highly correlated under mild conditions, but smaller variance alone does not guarantee higher accuracy. In the next section, we will empirically demonstrate that the new metric has a smaller variance and achieves higher accuracy. All proofs can be found in Appendix A.

## 5 EXPERIMENT RESULTS

### 5.1 ESTIMATION OF ACCURACY

Recall that accuracy of a metric $\bar{e}$ is defined as:

$$\text{Acc}(\bar{e}) \triangleq P(R_{\bar{e}}(h_A) < R_{\bar{e}}(h_B)). \tag{8}$$

To get an estimation of $\text{Acc}(\bar{e})$, we run pipelines $A$ and $B$ for $m$ times, obtaining models $h_{A_i}$ and $h_{B_i}$ for $i \in [m]$. $\text{Acc}(\bar{e})$ can be estimated as:

$$\widehat{\text{Acc}}(\bar{e}) = \frac{1}{m^2} \sum_{(i,j)} \mathbb{1}(\hat{R}_{\bar{e}}(h_{A_i}, \hat{\mathcal{D}}_{\text{test}}) < \hat{R}_{\bar{e}}(h_{B_j}, \hat{\mathcal{D}}_{\text{test}})) \tag{9}$$

$\widehat{\text{Acc}}(\bar{e})$ is an unbiased estimator of $\text{Acc}(\bar{e})$, and in the experiments below, we report $\widehat{\text{Acc}}(\bar{e})$ as our accuracy metric. In all the tables in this section, without loss of generality, we write the tables as pipeline A is better than pipeline B in the sense of $P(R_e(h_A) < R_e(h_B)) > 0.5$.

### 5.2 SYNTHETIC DATA

In Appendix B, we consider a linear regression model to give empirical evidence to support our theory. We further consider logistic regression model to demonstrate the effectiveness of Calibrated Log Loss in synthetic data setting. All the details and results can be found in Appendix B.

### 5.3 AVAZU CTR PREDICTION DATASET

**Dataset** The Avazu CTR Prediction dataset [1] is a common benchmark dataset for CTR predictions. Due to computational constraints in our experiments, we use the first 10 million samples, shuffle the dataset randomly, and split the whole dataset into $80\%\ \hat{\mathcal{D}}_{\text{train}}$, $2\%\ \hat{\mathcal{D}}_{\text{val}-\text{test}}$ and $18\%\ \hat{\mathcal{D}}_{\text{remaining}-\text{test}}$.

---

[1] https://www.kaggle.com/c/avazu-ctr-prediction

**Base Model** We use the xDeepFM model (Lian et al., 2018) open sourced in Shen (2017) as our base model. We primarily conduct experiments using xDeepFM models, including hyperparameter related experiments and feature related experiments. To demonstrate our new metric can also handle comparisons between different model architectures, we also conduct experiments using DCN (Wang et al., 2017), DeepFM (Guo et al., 2017b), FNN (Zhang et al., 2016), and DCNMix (Wang et al., 2021).

**Experiment Details** We consider neural networks with different architectures, different training methods, different hyper-parameters, and different levels of regularization as different pipelines. Such comparisons represent common practices for research and development in both industry and academia. For each pipeline, we train the model 60 times with different initialization seeds and data orders to calculate $\widehat{\text{Acc}}(\bar{e})$. Note that we use "Log Loss" as our ground truth metric to determine the performance rank of different pipelines. Due to computational constraints, we cannot afford to run the experiments for multiple rounds. Instead, we run the experiments for one round and report accuracy. Note that in the neural network experiments, we do not re-sample the training data each time, as there is intrinsic randomness in the neural network training process. This is the main difference from the Linear Regression and Logistic Regression experiments.

**Pipelines with Different Number of Features** In this set of experiments, for pipeline $A$, we use all the features available. For pipeline $B$, we remove some informative features. We tested the removal of 6 dense features and 1 sparse features respectively.

Table 1: Accuracy of Log Loss and Calibrated Log Loss under Neural Networks (features)

| Pipeline A | Pipeline B | Log Loss Acc | Calibrated Log Loss Acc |
|---|---|---|---|
| Baseline | remove dense | 81.8% | 88.8% |
| Baseline | remove sparse | 78.6% | 85.9% |

Table 2: Mean and Standard Deviation of Log Loss and Calibrated Log Loss under Neural Networks (features)

| Pipeline | Log Loss Mean | Calibrated Log Loss Mean | Log Loss Std | Calibrated Log Loss Std |
|---|---|---|---|---|
| dense | 0.37408 | 0.37403 | 0.00047 | 0.00038 |
| sparse | 0.37404 | 0.37398 | 0.0005 | 0.00042 |

From the result in Table 1, we can clearly see that Calibrated Log Loss has a higher accuracy, indicating its effectiveness when comparing the performance of pipelines with different features.

From the result in Table 2, we can see that Calibrated Log Loss has a smaller standard deviation (16% - 19% smaller) while the mean of Log Loss and Calibrated Log Loss is almost on par (within 0.02% difference).

**Pipelines with Different Model Architectures** In this set of experiments, for pipeline $A$, we use one model architecture. For pipeline $B$, we use another model architecture. We tested a variety of different model architectures, including DCN (Wang et al., 2017), DeepFM (Guo et al., 2017b), FNN (Zhang et al., 2016), and DCNMix (Wang et al., 2021).

Table 3: Accuracy of Log Loss and Calibrated Log Loss under Neural Networks (model architectures)

| Pipeline A | Pipeline B | Log Loss Acc | Calibrated Log Loss Acc |
|---|---|---|---|
| DCN | DCNMix | 64.4% | 71.5% |
| DeepFM | DCN | 77.2% | 83.9% |
| DeepFM | FNN | 76.9% | 79.9% |
| FNN | DCNMix | 61.5% | 72.0% |
| DeepFM | DCNMix | 84.8% | 93.4% |

From the result in Table 3, we can clearly see that Calibrated Log Loss has higher accuracy, again indicating its effectiveness when comparing the performance of pipelines with different model architectures. In Table 4, we report the mean and standard deviation of Log Loss and Calibrated Log Loss, consistent with previous results.

Table 4: Mean and Standard Deviation of Log Loss and Calibrated Log Loss under Neural Networks (model architectures)

| Pipeline | Log Loss Mean | Calibrated Log Loss Mean | Log Loss Std | Calibrated Log Loss Std |
|---|---|---|---|---|
| DCN | 0.38021 | 0.38011 | 0.00044 | 0.00033 |
| DeepFM | 0.37971 | 0.3796 | 0.00059 | 0.00037 |
| FNN | 0.38029 | 0.38006 | 0.00064 | 0.0004 |
| DCNMix | 0.38046 | 0.38037 | 0.00048 | 0.00034 |

**Pipelines with Different Model Hyperparameters** In this set of experiments, we compare pipelines with different model hyperparameters, including neural network layer size, Batch Normalization (BN) (Ioffe & Szegedy, 2015), Dropout (Srivastava et al., 2014), and regularization weight.

In the first experiment, we compare a pipeline using a baseline model size with a pipeline using a smaller model size. In the second experiment, we compare a pipeline using Batch Normalization with a pipeline not using Batch Normalization. In the third experiment, we compare a pipeline not using Dropout with a pipeline using Dropout with dropout probability 0.7. In the fourth experiment, we compare a pipeline not using regularization with a pipeline using L2 regularization with regularization weight $10^{-6}$.

Table 5: Accuracy of Log Loss and Calibrated Log Loss under Neural Networks (hyperparameters)

| Pipeline A | Pipeline B | Log Loss Acc | Calibrated Log Loss Acc |
|---|---|---|---|
| Baseline Size | Smaller Size | 69.6% | 73.6% |
| BN | no BN | 80.2% | 89.7% |
| no Dropout | p = 0.7 | 95.0% | 99.3% |
| no regularization | $10^{-6}$ | 95.2% | 98.8% |

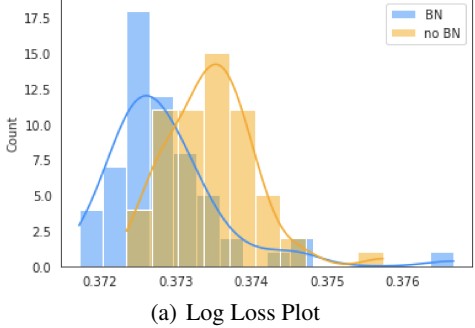

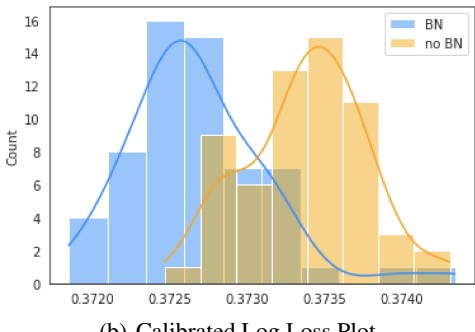

(a) Log Loss Plot          (b) Calibrated Log Loss Plot

Figure 1: Batch Normalization Experiment

From figure 1, we can see that by using Calibrated Log Loss, it becomes easier to separate pipeline using Batch Normalization from pipeline without Batch Normalization.

From the result in Table 5, we can see that Calibrated Log Loss has a higher accuracy regardless of the hyperparameters we are tuning, indicating its effectiveness when comparing the performance of pipelines with different hyperparameters, which is a very common task in Deep Learning. In Appendix B.3 Table 12, we report the mean and standard deviation of Log Loss and Calibrated Log Loss, again consistent with previous results.

**Pipelines with Different Levels of Regularization** In this set of experiments, we take a closer look at one hyperparameter we conduct in the previous section: regularization weight. For pipeline $A$, we use the baseline model. For pipeline $B$, we use different L2 regularization weights.

From the result in Table 6, we can see that Calibrated Log Loss has a higher accuracy across all different regularization weights, indicating its robustness to different values of regularization weight. As we increase the regularization weight in pipeline $B$, the accuracies of both metrics increase. This is because pipeline $A$ and $B$ differ more with larger regularization weight, making performance comparison easier.

Table 6: Accuracy of Log Loss and Calibrated Log Loss under Neural Networks (regularization weight)

| Pipeline A | Pipeline B | Log Loss Acc | Calibrated Log Loss Acc |
|---|---|---|---|
| 0 | $3 * 10^{-7}$ | 63.2% | 69.3% |
| 0 | $5 * 10^{-7}$ | 82.2% | 88.2% |
| 0 | $7 * 10^{-7}$ | 86.6% | 92.4% |
| 0 | $10^{-6}$ | 95.2% | 98.8% |
| 0 | $2 * 10^{-6}$ | 98.8% | 100.0% |

From the result in Appendix B.3 Table 13, we can see that Calibrated Log Loss has a much smaller standard deviation (15% - 40% smaller) than Log Loss while the mean of Log Loss and Calibrated Log Loss is almost on par (within 0.05% difference), again consistent with previous results.

## 5.4 CIFAR10

**Dataset** We use the default CIFAR10 train dataset as $\hat{\mathcal{D}}_{\text{train}}$. We split CIFAR10 test dataset into 20% $\hat{\mathcal{D}}_{\text{val-test}}$ and 80% $\hat{\mathcal{D}}_{\text{remaining-test}}$.

**Base Model** We use the ResNet18 model (He et al., 2016) as our base model. We compare ResNet18 with other configurations of ResNet and DenseNet (Huang et al., 2017). We used the open source implementations [2] to conduct all the experiments.

**Experiment Details** Same as Avazu CTR experiments, for each pipeline, we train the model 60 times with different initialization seeds and data orders to calculate $\widehat{\text{Acc}}(\bar{e})$. Note that we use "Classification Accuracy" as our ground truth metric to determine the performance rank of different pipelines. We train the model for 200 epochs. We run the experiments for one round and report accuracy. Note that in the neural network experiments, we do not re-sample the training data each time, as there is intrinsic randomness in the neural network training process.

**Calibration Stage** Predicting the image class in CIFAR10 is a Multiclass classification task, which is different from the binary classification task in the CTR prediction context. We use the "Temperature Scaling" (Guo et al., 2017a) as our calibration method.

**Experiment Results** From the result in Table 7, we can see that Calibrated Log Loss has a higher

Table 7: Accuracy of Log Loss and Calibrated Log Loss (CIFAR10)

| Pipeline A | Pipeline B | Log Loss Acc | Calibrated Log Loss Acc |
|---|---|---|---|
| resnet18 | resnet101 | 5.6% | 74.7% |
| resnet18 | resnet152 | 8.1% | 87.4% |
| resnet18 | DenseNet121 | 51.5% | 85.2% |
| resnet18 | resnet34 | 13.5% | 56.7% |

accuracy, indicating its effectiveness when comparing the performance of pipelines using different image classification models.

We can also see that sometimes the accuracy of Log Loss is below 50% (recall that Pipeline A is better than Pipeline B), which indicates that Log Loss and Classification Accuracy will give different results when comparing the performance of pipeline $A$ and pipeline $B$ (e.g. pipeline A performs better if using metric Log Loss while pipeline B performs better if using metric Classification Accuracy). This metric inconsistency can be mitigated by Calibrated Log Loss. This phenomenon indicates that some models may have bad calibration (and hence bad Log Loss) while their performances are actually good if measured by accuracy.

---

[2]https://github.com/kuangliu/pytorch-cifar

## 6    RELATED WORK

In recommendation systems, recommending proper items to users is a fundamental task. In order to do accurate recommendations, it is essential to build Click-Through Rate (CTR) prediction models to rank the items and achieve business goals. Prior to Deep Learning Era, traditional machine learning models (Friedman, 2001; Koren et al., 2009; Rendle, 2010; Desrosiers & Karypis, 2010; Canini et al., 2012) like logistic regression, boosted decision trees and factorization machines are commonly used to build CTR models. In the Deep Learning Era, not surprisingly, CTR prediction models have been transitioned to deep models as well (Cheng et al., 2016; Guo et al., 2017b; Covington et al., 2016; Wang et al., 2017; Zhou et al., 2018; Naumov et al., 2019).

A typical Deep Click-Through Rate (CTR) prediction model consists of embedding layers and multilayer perceptron (MLP) layers on top of embedding layers. Embedding layers transform raw discrete IDs to low dimensional vectors. Following embedding layers, MLP layers then learn the interactions of different features represented the low dimensional vectors, and finally output the final prediction (i.e. CTR). Currently, Deep Click-Through Rate (CTR) prediction models achieve state-of-the-art performance for CTR tasks, and are commonly used in industries to power various recommendation tasks, including personalized ads recommendations, content recommendations, etc.

There are a number of commonly used metrics (Yi et al., 2013) to evaluate the performance of CTR prediction models. Area Under the ROC Curve (AUC) (Fawcett, 2006; 2004) along with its variants Zhu et al. (2017) and Log Loss are the most common metrics. For example, He et al. (2014); Wang et al. (2017); McMahan et al. (2013) use Log Loss as their core metric, while Zhou et al. (2018); McMahan et al. (2013) use AUC as their core metric. However, AUC has been criticized for not taking into account the predictive probability (Yi et al., 2013). Log Loss, on the other hand, takes predictive probability into account. As a result, in applications where predictions are expected to be calibrated (e.g. Ads Recommendation Systems), Log Loss is preferred.

## 7    CONCLUSION AND DISCUSSION

**Conclusion**

In this paper, we have presented a new approach to comparing the performance of different deep learning pipelines. We proposed a new metric framework, Calibrated Loss, which has a higher accuracy and smaller variance than its vanilla counterpart for a wide range of pipelines. Our experiments in section 5 demonstrated the superiority of Calibrated Loss, and we believe this new metric can be used to more effectively and efficiently compare the performance of different pipelines in similar settings. Future work includes expanding this idea to evaluate NLP pipelines, and establish theoretical guarantees under more general settings.

**Limitations**

Our method sacrifices accuracy when comparing some specific pipelines. For example, if pipeline $B$ can reliably improve the model calibration in test distribution over pipeline $A$, Calibrated Log Loss will not be able to correctly detect the benefits of pipeline B, while Log Loss is able to. However, for most pipeline comparisons conducted in industry and academia like feature engineering, tuning parameters, etc., Calibrated Log Loss has a huge accuracy boost over Log Loss as we demonstrated in Section 5.

**Potential Applications**

Our method may have applications in AutoML domain. AutoML (Automated Machine Learning) systems are designed to automate the process of selecting, designing, and tuning machine learning models, and a key component of these systems is the selection of the best-performing pipeline (e.g. hyperparameters, model architectures etc.). The new metric can be used as a more accurate way of comparing the performance and selecting the best one. The new metric is in particular useful when performing hyperparameter tuning.

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

## A PROOFS

### A.1 PROOFS OF THEOREM 4.1 AND COROLLARY 4.2

**Lemma A.1.** *Suppose that $\hat{\beta}_n$ is the unique linear regression solution computed using the training data $\{X_i, Y_i\}_{i=1}^n$. Then, $\hat{\beta}_n$ is independent to $\{\bar{X}, \bar{Y}\}$, where*

$$\bar{X} = \frac{1}{n}\sum_{i=1}^n X_i, \text{ and } \bar{Y} = \frac{1}{n}\sum_{i=1}^n Y_i.$$

*Proof.* It is well-known that $\hat{\beta}_n$ is the solution of the convex program

$$\min_{\beta,c} \sum_{i=1}^{n} \left(Y_i - \beta^\top X_i - c\right)^2,$$

which is equivalent to the convex program

$$\min_{\beta} \sum_{i=1}^{n} \left(Y_i - \beta^\top X_i - \left(\bar{Y} - \beta^\top \bar{X}\right)\right)^2 \tag{10}$$

$$= \min_{\beta} \sum_{i=1}^{n} \left(\left(Y_i - \bar{Y}\right) - \beta^\top \left(X_i - \bar{X}\right)\right)^2.$$

Let $\tilde{Y}_i = Y_i - \bar{Y}$ and $\tilde{X}_i = Y_i - \bar{Y}$. Note that $\tilde{Y}_i$ is independent to $\bar{Y}$ and $\tilde{X}_i$ is independent to $\bar{X}$ as

$$cov(Y_i - \bar{Y}, \bar{Y}) = 0, cov(X_i - \bar{X}, \bar{X}) = 0,$$

and $\{X, Y\}$ are jointly normal. Note that the convex program ( 10) yields that $\hat{\beta}_n$ is a function of $\left\{\tilde{X}_i, \tilde{Y}_i\right\}_{i=1}^{n}$, which is independent to $\left\{\bar{X}, \bar{Y}\right\}$. $\square$

**Theorem A.2.** *Suppose that the features $X \in \mathbb{R}^d$ and the label $Y$ are distributed jointly Gaussian. We consider linear regression $h(x) = \beta^\top x + \alpha$. Let $\hat{\beta}_n$ be the coefficient learned from the training data with sample size $n$. Then, we have*

$$\left(1 + \frac{1}{n}\right) \mathbb{E}[e_1(h(X), Y)|\hat{\beta}_n] = \mathbb{E}[e(h(X), Y)|\hat{\beta}_n],$$

*where the expectation is taken over the randomness over both the training and test samples.*

*Proof.* Note that the learned bias $\hat{\alpha} = \bar{Y} - \hat{\beta}_n^\intercal \bar{X}$, where $\bar{Y}$ and $\bar{X}$ are the empirical average of the samples in the training set. Then, The risks are defined as

$$\mathbb{E}\left[e(h(X), Y)|\hat{\beta}_n\right] = \mathbb{E}\left[\left(\left(Y - \bar{Y}\right) - \hat{\beta}_n^\intercal \left(X - \bar{X}\right)\right)^2 |\hat{\beta}_n\right],$$

$$\mathbb{E}\left[e_1(h(X), Y)|\hat{\beta}_n\right] = \mathbb{E}\left[\left(\left(Y - \mathbb{E}_{\mathcal{D}}[Y]\right) - \hat{\beta}_n^\intercal \left(X - \mathbb{E}_{\mathcal{D}}[X]\right)\right)^2 |\hat{\beta}_n\right].$$

Therefore, we have

$$\mathbb{E}\left[e_1(h(X), Y)|\hat{\beta}_n\right] = var(Y - \hat{\beta}_n^\intercal X|\hat{\beta}_n).$$

Note that $\hat{\beta}_n$ is independent to $\left\{\bar{X}, \bar{Y}\right\}$, we have

$$\{Y - \bar{Y}, X - \bar{X}\} \overset{d}{=} \sqrt{\frac{n+1}{n}} \{Y, X\}.$$

Therefore, we have

$$\mathbb{E}\left[e(h(X), Y)|\hat{\beta}_n\right] = var\left(\left(Y - \bar{Y}\right) - \hat{\beta}_n^\intercal \left(X - \bar{X}\right) |\hat{\beta}_n\right)$$

$$= \left(1 + \frac{1}{n}\right) var(Y - \hat{\beta}_n^\intercal X|\hat{\beta}_n).$$

$\square$

**Corollary A.3.** *Suppose that $h_1(x)$ and $h_2(x)$ are two different learned linear functions in different feature sets. Then, we have*

$$\mathbb{E}[R_e(h_1)] = \mathbb{E}[R_e(h_2)] \Leftrightarrow \mathbb{E}[R_{e_1}(h_1)] = \mathbb{E}[R_{e_1}(h_2)] \tag{11}$$

*and $var((1 + 1/n) R_{e_1}(h)) < var(R_e(h))$ for any $h$ learned from linear regression.*

*Proof.* From the definition, we see

$$
\begin{aligned}
\mathbb{E}[R_e(h)] &= \mathbb{E}\left[\mathbb{E}\left[e(h(X),Y)|\hat{\beta}_n\right]\right], \\
\mathbb{E}[R_{e_1}(h)] &= \mathbb{E}\left[\mathbb{E}\left[e_1(h(X),Y)|\hat{\beta}_n\right]\right].
\end{aligned}
$$

Therefore, we conclude the first claim.

For the second claim, note that

$$
\begin{aligned}
R_{e_1}(h) &= \mathbb{E}\left[e_1(h(X),Y)|\hat{\beta}_n\right], \\
R_e(h) &= \mathbb{E}\left[e_1(h(X),Y)|\hat{\beta}_n,\bar{X},\bar{Y}\right].
\end{aligned}
$$

Then, the variance of $R_e(h)$ can be decomposed as

$$
\begin{aligned}
var(R_e(h)) &= var\left(\mathbb{E}\left[e(h(X),Y)|\hat{\beta}_n\right]\right) + \mathbb{E}\left[var\left(\mathbb{E}\left[e(h(X),Y)|\hat{\beta}_n^{\mathsf{T}},\bar{X},\bar{Y}\right]|\hat{\beta}_n\right)\right] \\
&> var\left(\mathbb{E}\left[e(h(X),Y)|\hat{\beta}_n\right]\right) \\
&= var\left(\left(1+\frac{1}{n}\right)\mathbb{E}\left[e_1(h(X),Y)|\hat{\beta}_n\right]\right) \\
&= var\left(\left(1+\frac{1}{n}\right)R_{e_1}(h)\right).
\end{aligned}
$$

$\square$

# B EXPERIMENTS

## B.1 SYNTHETIC DATA: LINEAR REGRESSION

We consider a linear regression model in this section to give empirical evidence to support our theory. We assume the response $Y$ follows the following generating process:

$$
Y = \beta^\top X + \epsilon, \tag{12}
$$

where $\epsilon \sim \mathcal{N}(\mu_e, \Sigma_e)$ and $\beta, X \in \mathbb{R}^d$.

In the experiments, we consider $d = 20$, $\beta = [1, 1, \ldots, 1]^\top$, and $X \sim \mathcal{N}(\mu_{\mathcal{D}}, \Sigma_{\mathcal{D}})$ in both the training set and the test set. In the training set, we generate $N_{\text{train}} = 1000$ i.i.d. training samples to train a linear regression model. In the test set, we generate $N_{\text{test}} = 11000$ i.i.d. test samples, with $N_{\text{val}-\text{test}} = 1000$ and $N_{\text{remaining}-\text{test}} = 10000$.

We assume $\mu_{\mathcal{D}} = [-0.05, -0.05, \ldots, -0.05]^\top$, $\Sigma_{\mathcal{D}} = 0.25^2 \times I_{d \times d}$, $\mu_e = 1$ and $\Sigma_e = 2$.

Note that there is no randomness in the training process of Linear Regression, as it's a convex optimization program. The randomness of Linear Regression comes from the training data. In order to run pipelines $A$ and $B$ multiple times to estimate the metric accuracy, we re-sample training data each time from the ground truth data distribution.

For pipeline $A$, we use all the 20 features available, and for pipeline $B$, we use the first 19 features and leave the last feature out. It's clear that pipeline $A$ should perform better than pipeline $B$ in the ground truth.

For each round of experiments, we run pipelines A and B $m = 100$ times and report $\widehat{\text{Acc}}(\bar{e})$, shown as "Acc" in the table 8. We performed 20 rounds of experiments, and report the mean and the standard errors of $\widehat{\text{Acc}}(\bar{e})$ in Table 8. We also calculate the standard deviation and mean of Quadratic Loss and Calibrated Quadratic Loss from pipeline A in each round of experiments, and report the average in Table 9.

From the result in Table 8, we can see that Calibrated Quadratic Loss has a higher accuracy compared with Quadratic Loss. From the result in Table 9, we can see that Calibrated Quadratic Loss indeed has a smaller standard deviation (3.1% smaller) than Quadratic Loss while the mean of Quadratic Loss and Calibrated Quadratic Loss is almost on par (0.07% difference).

Table 8: Accuracy of Quadratic Loss and Calibrated Quadratic Loss under Linear Regression

| Pipeline A | Pipeline B | Quadratic Loss Acc | Calibrated Quadratic Loss Acc |
|---|---|---|---|
| 20 features | 19 features | $93.49\% \pm 0.35\%$ | $95.81\% \pm 0.28\%$ |

Table 9: Mean and Standard Deviation of Quadratic Loss and Calibrated Quadratic Loss under Linear Regression

| Quad Loss Mean | Calibrated Quadratic Loss Mean | Quad Loss Std | Calibrated Quadratic Loss Std |
|---|---|---|---|
| 4.067 | 4.070 | 0.0295 | 0.0286 |

## B.2 SYNTHETIC DATA: LOGISTIC REGRESSION

We consider a logistic regression model. We assume the response $Y$ follows the Bernoulli distribution with probability $\left(1 + \exp(-\beta^\top X)\right)^{-1}$, for $\beta, X \in \mathbb{R}^d$.

In the experiments, we consider $d = 20$, $\beta = [1, 1, \ldots, 1]^\top$, and $X \sim \mathcal{N}(\mu_\mathcal{D}, \Sigma_\mathcal{D})$ in both the training and test sets. In the training set, we generate $N_{\text{train}} = 1000$ i.i.d. training samples to train a logistic regression model. In the test set, we generate $N_{\text{test}} = 12000$ i.i.d. test samples, with $N_{\text{val}-\text{test}} = 2000$ and $N_{\text{remaining}-\text{test}} = 10000$ .

We assume $\mu_\mathcal{D} = [-0.05, -0.05, \ldots, -0.05]^\top$ and $\Sigma_\mathcal{D} = 0.25^2 \times I_{d \times d}$.

Note that similar to Linear Regression, there is no randomness in the training process of Logistic Regression as well, as it's a convex optimization program. The randomness of Logistic Regression comes from the training data. We employ the same strategy to estimate the metric accuracy, i.e. we re-sample training data each time from the ground truth data distribution.

For pipeline $A$, we use all the 20 features available, and for pipeline $B$, we use the first 19 features and leave the last feature out. It's clear that pipeline $A$ should perform better than pipeline $B$ in the ground truth.

For each round of experiments, we run pipelines A and B $m = 1000$ times and report $\widehat{\text{Acc}}(\bar{e})$, shown as "Acc" in the table 10. We performed 20 rounds of experiments, and report the mean and the standard errors of $\widehat{\text{Acc}}(\bar{e})$ in Table 10. We also calculate the standard deviation and mean of Log Loss and Calibrated Log Loss from pipeline A in each round of experiments, and report the average in Table 11.

Table 10: Accuracy of Log Loss and Calibrated Log Loss under Logistic Regression

| Pipeline A | Pipeline B | Log Loss Acc | Calibrated Log Loss Acc |
|---|---|---|---|
| 20 features | 19 features | $79.62\% \pm 0.18\%$ | $83.7\% \pm 0.15\%$ |

From the result in Table 10, we can clearly see that Calibrated Log Loss has a huge accuracy boost compared with Log Loss. From the result in Table 11, we can see that Calibrated Log Loss indeed has a smaller standard deviation (3.9% smaller) than Log Loss while the mean of Log Loss and Calibrated Log Loss is almost on par (0.43% difference).

## B.3 AVAZU CTR PREDICTION DATASET

We report the mean and standard deviation of Log Loss and Calibrated Log Loss for additional Avazu CTR Prediction dataset experiments.

Table 11: Mean and Standard Deviation of Log Loss and Calibrated Log Loss under Logistic Regression

| Log Loss Mean | Calibrated Log Loss Mean | Log Loss Std | Calibrated Log Loss Std |
|---|---|---|---|
| 0.5366 | 0.5343 | 0.00385 | 0.00370 |

Table 12: Mean and Standard Deviation of Log Loss and Calibrated Log Loss under Neural Networks (hyperparameters)

| Pipeline | Log Loss Mean | Calibrated Log Loss Mean | Log Loss Std | Calibrated Log Loss Std |
|---|---|---|---|---|
| Baseline | 0.37347 | 0.37338 | 0.00058 | 0.00038 |
| smaller size | 0.37383 | 0.37374 | 0.00048 | 0.00039 |
| BN | 0.37286 | 0.37268 | 0.00078 | 0.00044 |
| Dropout | 0.37454 | 0.37456 | 0.00039 | 0.00036 |
| Regularization | 0.37475 | 0.37459 | 0.00061 | 0.00042 |

Table 13: Mean and Standard Deviation of Log Loss and Calibrated Log Loss under Neural Networks (regularization weight)

| Pipeline | Log Loss Mean | Calibrated Log Loss Mean | Log Loss Std | Calibrated Log Loss Std |
|---|---|---|---|---|
| 0 | 0.37347 | 0.37338 | 0.00058 | 0.00038 |
| 3e-7 | 0.37371 | 0.37369 | 0.00048 | 0.00043 |
| 5e-7 | 0.37419 | 0.37411 | 0.00059 | 0.0005 |
| 7e-7 | 0.37428 | 0.37421 | 0.00057 | 0.00044 |
| 1e-6 | 0.37475 | 0.37459 | 0.00061 | 0.00042 |
| 2e-6 | 0.37562 | 0.37547 | 0.00059 | 0.00042 |

