# OpenReview forum: "Calibration-then-Calculation: A Variance Reduced Metric Framework"
_ICLR.cc/2024/Conference — Submitted to ICLR 2024_

### Official Review · Reviewer_HhCS · 2023-10-20

**Soundness:** 3 good
**Presentation:** 3 good
**Contribution:** 3 good
**Rating:** 6
**Confidence:** 4

**Summary:**

This paper presented a new approach to comparing the performance of different deep learning pipelines and proposed a new metric framework, Calibrated Loss, which has a higher accuracy and smaller variance than its vanilla counterpart for a wide range of pipelines.

**Strengths:**

A new evaluation framework which can save much computation resource when it face accurate or multiple evaluations.

**Weaknesses:**

The generalization part in section 3 is vague. It is better to illustrate how to generalise in detail.

**Questions:**

1. I don't understand the first line of Table 7 shows the log loss acc is only 5.6% for resnet101 over resnet18. Could you please explain more?
2. Log loss can sometime indicate the classification accuracy of a model, but they are not identical. How you define model A is better than model B? If I want to focus on accuracy, not log loss, how to apply your method?

---

> ### Author Response · Authors · 2023-11-20
>
> We would like to express our gratitude to the reviewer for their thorough feedback and valuable suggestions. We have carefully examined the comments and we present our responses to the reviewer's concerns below:
>
> Q1: response to question 1
>
> A1: We discussed the reason at the bottom of page 8. It indicates that Log Loss and Classification Accuracy will give different results when comparing the performance of pipeline A and pipeline B (e.g. pipeline A performs better if using metric Log Loss while pipeline B performs better if using metric Classification Accuracy). This metric inconsistency can be mitigated by Calibrated Log Loss. This phenomenon indicates that some models may have bad calibration (and hence bad Log Loss) while their performances are actually good if measured by accuracy.
>
> Q2: response to question 2
>
> A2: Thanks for the great question. The definition of “better” is at page 2, equestion 3. If only the accuracy metric is of interest, then the proposed method is not applicable as calibration won’t change accuracy.

---

### Official Review · Reviewer_cepe · 2023-11-06

**Soundness:** 2 fair
**Presentation:** 2 fair
**Contribution:** 2 fair
**Rating:** 3
**Confidence:** 4

**Summary:**

This paper proposes a new metric, Calibrated Loss, that addresses the challenges of evaluating deep learning pipelines. Calibrated Loss is a variance-reduced version of the vanilla loss metric, which makes it more accurate in detecting effective modeling improvement.

The authors provide theoretical justifications for their approach and validate it on two different types of deep learning models: Click Through Rate Prediction and Image Classification models.

**Strengths:**

1. Effective evaluation is critical for the development of machine learning algorithms.
2. The paper is well-organized and easy to follow.
3. Implementation is provided in the supplementary material.

**Weaknesses:**

### Major issues
1. The notations are unnecessarily redundant. For example, we can use $e, f$ to represent the metrics, and $A, B$ to represent the pipelines. $h$ represents the model name. It is redundant since it is a subset of the pipeline. Please reduce the usage of necessary superscripts and subscripts.
2. The paper focuses on the comparisons between two pipelines, which was not necessary. The paper should pay attention to the variance reduction in the performance evaluation of pipelines, i.e., the equation at the bottom of Page 2. For example, the histograms in Figure 1 are good visualization for the variance reduction. Please add similar figures for other experiments.
3. How do we handle the bias-variance tradeoff in the equation at the bottom of Page 2?
4. The Calibrated Loss for multiclass classification is the existing method of "Temperature Scaling". Why is it a proposed method? What is the contribution on that in this paper?
5. Calibration of the evaluation results is a general topic. The paper lacks discussions and comparisons of the related work.
6. In Section 4, there is less randomness with a larger test dataset. The original metric $e$ is good enough for evaluation. The difference between $e_1$ and $e$ will become smaller with a larger $n$. However, for the $e_1$, we may need to split the test dataset into two disjoint subsets, one for calibration and one for true evaluation, which will decrease the effective dataset size $n$. In that case, can we claim that $e_1$ is better than $e$? If $n=2$, $e$ can use the average of these two samples to obtain the final result, while $e_1$ should use one example for calibration and use the other one as the final result. In this case, which case is better? What if $n=10k$ and even infinity?

### Minor issues
1. Please number every equation in the paper.
2. In Equation 3, $R_e(h)$ is a random variable. A is better than B iff $P(R_e(h_A) < R_e(h_B)) > 0.5$. However, we usually use expectation or its estimations to compare A and B. Specifically, A is better than B iff $\mathbb{E}(R_e(h_A)) < \mathbb{E}(R_e(h_B))$. For example, we mainly compare the average classification accuracy. Equation 3 seems uncommon, which may require further discussions and citations.
3. The theoretical justification is only on the linear regression, and the randomness only comes from the dataset, which is limited. A more general setting (e.g., convex problem, randomness in SGD) is expected.

**Questions:**

Please see the weaknesses for details.

---

> ### Author Response · Authors · 2023-11-20
>
> We would like to express our gratitude to the reviewer for their thorough feedback and valuable suggestions. We have carefully examined the comments, and we present our responses to the reviewer's concerns below:
>
> Q1: concern regarding notations
>
> A1: Thanks for the suggestion. We will fix it in the updated version of this paper.
>
> Q2: concern regarding “comparisons between two pipelines”
>
> A2: We believe “comparisons between two pipelines” is necessary and important. Simply doing variance reduction is not useful. As an extreme case, assigning 0 as the metric number will have zero variance, but it’s not useful at all. The goal of this paper is to introduce a new metric to compare pipelines more accurately. Consider a potential application of this new metric — AutoML. For each AutoML search, we need to determine which configuration is better. Due to metric randomness, there might be some noises in the AutoML search process. Using the new metric could improve the accuracy of search, and ideally the final model would be better. We haven’t conducted the AutoML experiments yet, and this is a planned follow-up work.
>
> This also answers the next question about “bias-variance tradeoff”: we evaluate the metric based on its accuracy, defined in Equation 5.
>
> Q3: concern regarding “Calibrated Loss for multiclass classification”
>
> A3: We demonstrated that the new metric has a higher accuracy compared with its vanilla counterpart. This demonstrates that our framework “first calibrate, then calculate metric number” is a general method that can be applied to different applications.
>
> Q4: concern regarding “Calibration of the evaluation results”
>
> A4: We are aware of the existing work on model calibrations, but to our knowledge we are the first to tackle the deep learning pipeline evaluation problem from the calibration perspective. Our main contribution is to propose that the strategy “first calibrate, then calculate metric number” improves the metric accuracy.
>
> Q5: concern regarding “less randomness with a larger test dataset”
>
> A5: From the pure statistical perspective, it is correct that large n will create less randomness. As you mentioned, there is a gap between theory and experiments. In the experiments with NN, randomness from SGD plays a more important role, so the theory may not directly apply there. In the SGD case, batch size may serve the role of n. Our theory serves as supporting evidence to our framework that under some simple circumstances (e.g. linear regression), our framework can reduce the variance in theory. We support our framework under more complex settings via experiments.

---

> > ### Comment · Reviewer_cepe · 2023-11-23
> > **Thanks for the response.**
> >
> > I really appreciate the authors' response. I have one question on the variance reduction.
> >
> > Let A be a random variable representing the measured performance on a pipeline. The randomness is in the pipeline inherently, such as the SGD. The authors propose a method B=f(A) such that E(B) = E(A) and var(B) < var(A). The authors mention an extreme case B = f(A) = 0, where var(B) = 0 but E(B) and E(A) are different. The informative metric is an unbiased (or small enough bias) one with reduced variance.

---

> > > ### Author Response · Authors · 2023-11-23
> > >
> > > Thanks for your response.
> > >
> > > Sorry it's not very clear to us what your question is. Could you elaborate on it?

---

### Official Review · Reviewer_hPPi · 2023-11-08

**Soundness:** 2 fair
**Presentation:** 3 good
**Contribution:** 2 fair
**Rating:** 5
**Confidence:** 4

**Summary:**

The authors use the validation set loss (in Algorithm 1) to recalibrate their model for the  quadratic loss, the binary logistic loss (with a additional intercept tuned on the validation set) and multi-class logistic losses (with a temperature parameter).
For the quadratic loss in linear regression, they prove that this procedure leads to variance reduction and verify that via synthetic experiments.
There is no such mathematical proof for the binary or the multi-class logistic loss, but the authors demonstrate variance reduction via a series of experiments on synthetic, CTR and image datasets.

**Strengths:**

1.
The large number of experiments with sparse and dense features, synthetic, CTR and image datasets seem to illustrate the basic idea of the paper on re-calibration quite well.

2.
Corollary 4.2 regarding linear regression seems to be an interesting new result.

**Weaknesses:**

1.
Since the log loss accuracy reported in Tables 1 - 7 does not use the validation test, whereas the calibrated log loss accuracy uses this validation set, it is no surprise that the calibrated log loss accuracy is always higher than the log loss accuracy.
This is similar to comparing a model whose hyperparameters are tuned via cross-validation with one that only sees the training set and does not undergo hyperparameter tuning via cross-validation, with the only difference being that the additional hyperparameter introduced in this method is separately introduced via a connection to calibration.
The variance reduction property should also not be considered a surprise if the baseline is overfit on the training set.

2.
The lack of theoretical guarantees on the binary or the multi-class logistic loss also makes one wonder if the variance reduction property holds in general.

3.
Typos exist in the paper, e.g., "Calibratied" after Corollary 4.2

**Questions:**

Is the first pipeline in Table 2 labeled "dense" the same as the first pipeline B in Table 1 labled "remove dense" ?
If not, the reason for removing dense in Table 1's first row and keeping dense in Table 2's first row is not clear.

---

> ### Author Response · Authors · 2023-11-20
>
> We would like to express our gratitude to the reviewer for their thorough feedback and valuable suggestions. We have carefully examined the comments, and it seems there is a **misunderstanding** about this paper. We present our responses to the reviewer's concerns below:
>
> Q1: response to Weakness 1
>
> A1: There is a misunderstanding here. When calculating the vanilla log loss, we apply D_test = D_val-test + D_remaining-test. We intend to revise our paper to emphasize this crucial setting, and we're grateful for your attention to this detail.
>
> Q2: response to the question
>
> A2: Thanks for the great suggestion and spotting the typo. They are the same, and we should’ve used the same label “remove dense”. We will fix it in the updated version.

---

> ### Comment · Reviewer_hPPi · 2023-11-23
> **Thank you for the rebuttal !**
>
> I don't think there is a misunderstanding in Q1, but my original sentence was not precise enough. My original sentence should have been:
> Since the log loss accuracy reported in Tables 1 - 7 does not benefit from the use the validation test to tune additional calibration hyper-parameters, whereas the calibrated log loss accuracy uses this validation set, it is no surprise that the calibrated log loss accuracy is always higher than the log loss accuracy.
>
> I would like to retain my original rating given the concerns from other reviewers as well as the rest of my original concern in Q1 (which clearly conveyed the intent behind the first sentence):
> This is similar to comparing a model whose hyperparameters are tuned via cross-validation with one that only sees the training set and does not undergo hyperparameter tuning via cross-validation, with the only difference being that the additional hyperparameter introduced in this method is separately introduced via a connection to calibration. The variance reduction property should also not be considered a surprise if the baseline is overfit on the training set.
>
> Whether one reports the metrics on $ D_{val-test} + D_{remaining-test}$ or $ D_{remaining-test}$, when comparing methods, is less important. What is materially important is that the proposed method gets an unfair peek into $ D_{val-test}$, when tuning the calibration hyper-parameter.

---

> > ### Author Response · Authors · 2023-11-23
> >
> > Thanks for your response.
> >
> > First, we shuffle the dataset randomly (Section 5.3), so the distribution of D_test, D_val-test, D_remaining-test are the same. This is to avoid "learning from D_val-test" contains the additional information about the test distribution.
> >
> > Second, the reduced variance and higher accuracy when comparing pipelines may not be trivial. Consider an extreme case that there is only 1 sample left in D_remaining-test. In this case, the metric variance will be very high, even though the proposed method still "gets an unfair peek into D_val-test". And this is not unfair --- we can think of each metric as a function of the test data (not the test-remaining set), and the model. The goal of a metric is to compare pipelines more accurately, no matter how to utilize the test data. The common wisdom is to compute the metric based on the whole test dataset, which this paper shows that this may not be the best way to use all the test data.

---

### Official Review · Reviewer_Rghz · 2023-11-08

**Soundness:** 2 fair
**Presentation:** 3 good
**Contribution:** 3 good
**Rating:** 6
**Confidence:** 3

**Summary:**

This paper provides a new framework to evaluate deep learning pipelines. For classification, the model output is adjusted by a posthoc calibration method on the test-validation dataset, then the adjusted model is evaluated on the remaining test set. For normal regression, the bias-adjusted term is calculated on the test-validation dataset, then the quadratic loss is applied to the bias-adjusted predictions on the remaining test set.
Theoretically, the paper shows the proposed metric has lower variance for linear regression. Empirically, on click through rate predictions and image classification data sets, the proposed metric has low variance and better accuracy.

**Strengths:**

Providing a better evaluation of the deep learning pipeline is an interesting topic. This paper provides an easy-to-understand and easy-to-implement framework to improve the metric variance and accuracy. Numerical examples demonstrate the proposed metrics provide a better way to compare different deep learning pipelines.

**Weaknesses:**

1. My understanding is that the proposed framework applied a posthoc calibration on the trained model using a calibration set to adjust model prediction, then used the same loss function(cross-entropy loss for classification and $L_2$ loss for regression) to the adjusted model output on a separate test set. Frame it as a new loss is a bit confusing

2. The $\text{Acc}(\bar{e})$ defined in equation 5 relies on a ground truth metric, which makes the framework a bit hard to interpret. For example, in the CIFAR10 experiment, accuracy and log loss give different comparison results, I think this can happen because the larger model overfits in terms of cross-entropy loss, but accuracy doesn't drop(e.g. Figure 3 in [1]). The larger model is more accurate but not well-calibrated[1]. In the paper, it says "this metric inconsistency can be mitigated by Calibrated Log Loss", but I think a better way to explain the result is that after applying the posthoc calibration method, the larger model can have better calibration(in terms of cross-entropy loss).

3. Some subtle issues:

(1) accuracy is used as the ground truth metric in CIFAR10 experiments, but a model with higher accuracy or lower loss is better, this can cause some problems in the definition 3.

(2) On top of page 4, it says the bias-adjusted predictions $q_i$ are well-calibrated. But calibration usually means $E(Y|f(X)) = f(X)$ for all $f(X)$, the property only says the calibration error is 0 if evaluated using only 1 bin. So I think the claim is not appropriate.

(3) For the Generalization to Quadratic Loss, $e_1$ is defined using $E_{D}$, I think $E_{D}$ needs to be estimated on the val-test set, would it be better to define $e_1$ using $E_{\hat{D}_{\text{val-test}}}$?


### Reference
[1] Guo, Chuan, et al. "On calibration of modern neural networks." International conference on machine learning. PMLR, 2017.

**Questions:**

Please see the weakness part.

An additional question, in several comparison results shown in the paper, although the "Calibrated Log Loss Acc" is larger, it is still around 60% or 70%, far from the significance level usually used in statistical hypothesis testing to compare two methods. When we have these comparison accuracies, what would the authors suggest to say about the comparison of the two pipelines?

---

> ### Author Response · Authors · 2023-11-20
>
> We would like to express our gratitude to the reviewer for their thorough feedback and valuable suggestions. We have carefully examined the comments, and we present our responses to the reviewer's concerns below:
>
> Q1: Concern regarding framing as a new loss
>
> A1: Thanks for the great suggestions. We agree that the term “loss” is a bit confusing. We will change the framing to “a new metric”.
>
> Q2: Concern regarding the interpretation of the framework
>
> A2: Thanks for the great question. In real world applications, models are built to achieve some business needs. In different contexts, the best model may not be the same. For example, if the models are only used to classify images, then accuracy should be the main metric. If the models need to output the confidence, then calibration should be taken into account. That’s the main reason why we need a ground truth metric. This ground truth metric represents the business goal of the models.
>
> Q3: Concern regarding definition 3
>
> A3: Thanks for the great suggestion. In this case, we can use minus Accuracy as “risk”. We will make it clear in the paper.
>
> Q4: Concern regarding “well-calibrated” on the page of top 4
>
> A4: Thanks for the great suggestion. We will fix it in our next version of the paper.
>
> Q5: Concern regarding the subscript in section “Generalization to Quadratic Loss”
>
> A5: Thanks for the great suggestion. We will fix it in our next version of the paper.
>
> Q6: Concern regarding “statistical hypothesis testing to compare two methods”
>
> A6: Thanks for the great question. In real world applications, we cannot afford to run the pipeline multiple times. The goal of this paper is to introduce a new metric to compare pipelines more accurately, but we cannot guarantee 100% accuracy. Consider a potential application of this new metric — AutoML. For each AutoML search, we need to determine which configuration is better. Due to metric randomness, there might be some noises in the AutoML search process. Using the new metric could improve the accuracy of search, and ideally the final model would be better. We haven’t conducted the AutoML experiments yet, and this is a planned follow-up work.

---

> > ### Comment · Reviewer_Rghz · 2023-11-22
> >
> > Thank you for the detailed response. I will keep my rating

---

### Official Review · Reviewer_mk4m · 2023-11-09

**Soundness:** 3 good
**Presentation:** 3 good
**Contribution:** 2 fair
**Rating:** 3
**Confidence:** 3

**Summary:**

The submission proposes a new metric to evaluate deep learning models. The main idea is to calibrate the "bias" of the deep learning model that contributes to the randomness in evaluating them. The algorithm is proposed to split the test set into test-val and test-remaining splits and then using the test-val set to calibrate the model predictions before calculating the metric values. The accuracy of the proposed metric itself is also defined as the probability that a better model A will be measured as being better by the proposed metric.
The evaluation of the proposed metrics is done using deep CTR prediction and image classification.

**Strengths:**

- The idea of tackling randomness in evaluation of deep learning models is important.
- The proposed method seems to have strong theoretical background.
- The empirical evaluations are in favor of the proposed metric.

**Weaknesses:**

- I am not fully convinced that the proposed algorithm is realistic. For instance, to obtain the calibrated metric one must have a separate validation set, which is not always the case. One could split the test set into test-val and test-test splits like described in the submission, but that means using a different set of data to test the model, which in itself would be another problem. Furthermore, the calibrated metric seems to be model and data specific as it uses a specific model's predictions and validation data to calibrate the bias term. This would lead to evaluating multiple models which all use individually different metrics. I do not particularly think that would be a step towards robust and fair comparison of deep learning models.

**Questions:**

- As mentioned in the weakness, wouldn't the proposed algorithm lead to different metrics being used for different combinations of models and validation data?

---

> ### Author Response · Authors · 2023-11-20
>
> We would like to express our gratitude to the reviewer for the feedback. We have carefully examined the comments, and it seems there is a **misunderstanding** about this paper. We present our responses to the reviewer's concerns below:
>
> Q1: Concern regarding the proposed algorithm
>
> A1: We don’t need a separate validation set for the algorithm to work. We can always split test data into test-val and test-test splits as you mentioned. This test data split is deterministic, so every model will be evaluated on the same test-val data and test-test data.
>
> And could you elaborate on “the calibrated metric seems to be model and data specific”? Every metric is model specific and data specific — every metric needs to use model predictions on some data as input to calculate a number. At a high level, the metric is just a function of the model and data, and the function form could be general, and may not admit the expectation form, as we discussed in detail in Section 2 ” PRELIMINARIES AND PROBLEM SETTING”.

---

> > ### Comment · Reviewer_mk4m · 2023-11-23
> > **Reviewer response**
> >
> > >A1: We don’t need a separate validation set for the algorithm to work. We can always split test data into test-val and test-test splits as you mentioned. This test data split is deterministic, so every model will be evaluated on the same test-val data and test-test data.
> >
> > I meant that this would make comparison to existing benchmark results difficult, as previous metrics most likely used all the test data for benchmark purposes.
> >
> > >And could you elaborate on “the calibrated metric seems to be model and data specific”?
> >
> > I meant that the actual function that the metrics are calculated with may differ on a per model-data basis, which means even on the test-remaining set, different models may use different metrics for evaluation, which seems odd to me.

---

> > > ### Author Response · Authors · 2023-11-23
> > >
> > > Thank you for the response.
> > >
> > > 1. Indeed. Given we are proposing a new metric, the new metric result is not directly comparable with previous benchmark results. However, let's consider a potential application of this new metric — AutoML. For each AutoML search, we need to determine which configuration is better. Due to metric randomness, there might be some noises in the AutoML search process. Using the new metric could improve the accuracy of search, and ideally the final model would be better. We haven’t conducted the AutoML experiments yet, and this is a planned follow-up work.
> > >
> > > 2. We can think of each metric as a function of the test data (not the test-remaining set), and the model. The function associated with our new proposed metric is deterministic on the test data. The function is indeed not deterministic on the test-remaining set for different model, but that's not a problem. That's exactly the reason why the new proposed metric has the desired variance reduced property.

---

### Meta-Review · Area_Chair_7ZuJ · 2023-12-06

**Metareview:**

The paper introduces "Calibrated Loss," a new metric for evaluating deep learning models. It aims to address the randomness in model evaluations by recalibrating the bias in deep learning models using a split test set (test-val and test-remaining). The metric's accuracy is assessed through deep CTR prediction and image classification tests.

Strengths:
- The potential impact is significant, addressing the crucial issue of randomness in deep learning evaluations.

Weaknesses:
- The feasibility of the proposed algorithm in real-world scenarios is questionable due to the need for a separate validation set.
- The metric appears to be model and data-specific, potentially leading to inconsistent results across different models.
- Sections of the paper, such as the generalization part, are vaguely presented.

The paper, while addressing an important issue in deep learning evaluation, falls short in its practical applicability and theoretical comprehensiveness.

**Justification For Why Not Higher Score:**

N/A

**Justification For Why Not Lower Score:**

N/A

---

### Decision · Program_Chairs · 2024-01-16

Reject